# Epidemiology of soil transmitted helminths and risk analysis of hookworm infections in the community: Results from the DeWorm3 Trial in southern India

Sitara S. R. Ajjampur[1]*, Saravanakumar Puthupalayam Kaliappan[1], Katherine E. Halliday[2,3], Gokila Palanisamy[1], Jasmine Farzana[1], Malathi Manuel[1], Dilip Abraham[1], Selvi Laxmanan[1], Kumudha Aruldas[1], Anuradha Rose[4], David S. Kennedy[2], William E. Oswald[2], Rachel L. Pullan[2], Sean R. Galagan[3,5], Kristjana Ásbjörnsdóttir[3,6], Roy M. Anderson[7], Jayaprakash Muliyil[1], Rajiv Sarkar[8], Gagandeep Kang[1], Judd L. Walson[3,5,9]

1 The Wellcome Trust Research Laboratory, Division of Gastrointestinal Sciences, Christian Medical College, Vellore, India, 2 Department of Disease Control, Faculty of Infectious and Tropical Diseases, London School of Hygiene & Tropical Medicine, London, United Kingdom, 3 The DeWorm3 Project, University of Washington, Seattle, Washington, United States of America, 4 Department of Community Medicine, Christian Medical College, Vellore, India, 5 Department of Global Health, University of Washington, Seattle, Washington, United States of America, 6 Department of Epidemiology, University of Washington, Seattle, Washington, United States of America, 7 School of Public Health, Faculty of Medicine, Imperial College, London, United Kingdom, 8 Indian Institute of Public Health, Shillong, India, 9 Department of Medicine (Infectious Diseases) and Pediatrics, University of Washington, Seattle, Washington, United States of America

* sitararao@cmcvellore.ac.in

## Abstract

Since 2015, India has coordinated the largest school-based deworming program globally, targeting soil-transmitted helminths (STH) in ~250 million children aged 1 to 19 years twice yearly. Despite substantial progress in reduction of morbidity associated with STH, reinfection rates in endemic communities remain high. We conducted a community based parasitological survey in Tamil Nadu as part of the DeWorm3 Project—a cluster-randomised trial evaluating the feasibility of interrupting STH transmission at three geographically distinct sites in Africa and Asia—allowing the estimation of STH prevalence and analysis of associated factors. In India, following a comprehensive census, enumerating 140,932 individuals in 36,536 households along with geospatial mapping of households, an age-stratified sample of individuals was recruited into a longitudinal monitoring cohort (December 2017-February 2018) to be followed for five years. At enrolment, a total of 6089 consenting individuals across 40 study clusters provided a single adequate stool sample for analysis using the Kato-Katz method, as well as answering a questionnaire covering individual and household level factors. The unweighted STH prevalence was 17.0% (95% confidence interval [95% CI]: 16.0–17.9%), increasing to 21.4% when weighted by age and cluster size. Hookworm was the predominant species, with a weighted infection prevalence of 21.0%, the majority of which (92.9%) were light intensity infections. Factors associated with hookworm infection were modelled using mixed-effects multilevel logistic regression for presence of infection

(dw3data@uw.edu) for researchers who meet the criteria for access to these data.

**Funding:** The DeWorm3 study is funded through a grant to the Natural History Museum, London from the Bill and Melinda Gates Foundation (OPP1129535, PI JLW). SSRA is supported by an Emerging Global Leader Award (K43) from Fogarty International Center, NIH (1K43TW011415). The funders were not involved in the decision to publish the manuscript and had no role in data collection, analysis or publication of study results.

**Competing interests:** The authors have declared that no competing interests exist.

and mixed-effects negative binomial regression for intensity. The prevalence of both *Ascaris lumbricoides* and *Trichuris trichiura* infections were rare (<1%) and risk factors were therefore not assessed. Increasing age (multivariable odds ratio [mOR] 21.4, 95%CI: 12.3–37.2, p<0.001 for adult age-groups versus pre-school children) and higher vegetation were associated with an increased odds of hookworm infection, whereas recent deworming (mOR 0.3, 95%CI: 0.2–0.5, p<0.001) and belonging to households with higher socioeconomic status (mOR 0.3, 95%CI: 0.2–0.5, p<0.001) and higher education level of the household head (mOR 0.4, 95%CI: 0.3–0.6, p<0.001) were associated with lower odds of hookworm infection in the multilevel model. The same factors were associated with intensity of infection, with the use of improved sanitation facilities also correlated to lower infection intensities (multivariable infection intensity ratio [mIIR] 0.6, 95%CI: 0.4–0.9, p<0.016). Our findings suggest that a community-based approach is required to address the high hookworm burden in adults in this setting. Socioeconomic, education and sanitation improvements alongside mass drug administration would likely accelerate the drive to elimination in these communities.

**Trial Registration:** NCT03014167.

## Author summary

Approximately 1 in 5 people in India are infected with soil transmitted helminths (STH), leading to anaemia and malnutrition. To tackle this large burden of infection, the government of India launched one of the world's largest school-based deworming programs in 2015 aiming to deworm all pre-school and school-aged children between 1 to 19 years of age twice yearly on the National Deworming Days. Deworming programs, including those in India, are focused on pre-school aged children, school aged children and women of reproductive age group. However, prevailing environmental and socioeconomic conditions, including poor sanitation, can contribute to high rates of reinfection from untreated adults and children. The DeWorm3 Project is a cluster-randomised trial evaluating the feasibility of interrupting STH transmission with community wide deworming of all individuals aged one to 99 years of age or older. As part of the study, we conducted a parasitological survey in the Deworm3 trial site in rural Tamil Nadu. Here we present the factors associated with STH infection and burden in these communities.

## Introduction

Soil-transmitted helminths (STH)—*Ascaris lumbricoides*, hookworms (*Ancylostoma duodenale* and *Necator americanus*) and *Trichuris trichiura*—are among the most common infections globally, with India estimated to have the highest number of cases (375 million) according to the Global Burden of Disease estimates, 2013 [1]. Significant worldwide reductions in prevalence of *Ascaris* (-25.5% since 1990) have been estimated, but these reductions have been modest for *Trichuris* (-11.6%) and even smaller for hookworm (-5.1%) [2]. In more recent estimates (2015), 258 million (or 1 in 5) individuals in India are estimated to be infected with STH, with 148 million *Ascaris*, 109 million hookworm and 41 million *Trichuris* infections, indicating a lower prevalence of *Ascaris* and *Trichuris*, but a higher prevalence of hookworm than previous reports [3]. Moderate- and heavy-intensity (MHI) hookworm infections are

associated with lower haemoglobin levels and anaemia particularly affecting pregnant women and young children who often have low baseline iron stores [4–6]. While a recent Cochrane review indicated that regular deworming of children in public health programmes does not seem to improve outcomes [7], a study using data from Demographic and Health Surveys (DHS) of 45 STH endemic countries found that there was a consistent association between deworming and reduced stunting in pre-school-age children (PSAC) [8]. This is especially relevant in India, where more than half the children under 5 years are stunted [9]. Deworming has also been shown to improve nutritional status, cognition and school performance in school-age children (SAC) [10–12].

The WHO-recommended strategy is focused on controlling morbidity through mass drug administration (MDA) of anthelmintic drugs, albendazole or mebendazole, targeted to PSAC, SAC, women of reproductive age (WRA) and other at-risk populations, aiming for 75% coverage in these populations by 2020 [13,14]. Although the lymphatic filariasis (LF) control programme has delivered albendazole alongside diethylcarbamazine (DEC) through community-wide treatment in over 250 endemic districts in India since 2004 [15], STH burden has remained high [3,16]. The Ministry of Health and Family Welfare (MOHFW) in India has since introduced the world's largest school-based deworming program, targeting ~240 million children aged 1 to 19 years twice yearly (biannual) during the 'National Deworming Days' (NDD) conducted in February and August since 2015 [17]. Eleven states/union territories participated at the launch (including Tamil Nadu) and this program expanded to 33 states/union territories in 2019. With primary school enrolment exceeding 99% in India [18] and the involvement of anganwadi centres (a government run centre in each village providing care for pregnant women and children under 6 years of age under the Integrated Child Development Services Scheme), this is a highly effective way of reaching out to PSAC and SAC to carry out a targeted deworming program.

Reinfection rates in endemic communities with ongoing targeted deworming programs are often high due to poor sanitation, high rates of open defecation, migration and persistent reservoirs of infection in untreated adults [19]. While India has initiated large-scale programs to provide toilet access and reduce open defecation, [20] in the absence of significant structural improvements in sanitation, targeted deworming programs would likely need to be continued indefinitely [21]. Furthermore, meta-analyses, mathematical models and empirical field studies suggest that a community-wide deworming strategy including individuals of all ages may be effective in interrupting transmission of STH infections [22–24]. The recent launch of the WHO 2030 targets for STH control programmes has also emphasised the goal of achieving and maintaining elimination of STH morbidity in pre-SAC and SAC as well as the need to establish an efficient STH control programme for WRA. This has highlighted the need for robust epidemiological data to inform the strengthening of future efforts to control or interrupt transmission of STH [25].

The vast majority of empirical data monitoring the progress of the NDD and demonstrating STH burden reductions in children in India have been collected through school surveys [16,26]. To fully understand the current STH epidemiology, especially for hookworm infections known to increase and plateau in adulthood, community-wide data are also required [27]. We present the results of an age-stratified community STH survey conducted at baseline with participants enrolled into a longitudinal monitoring cohort as part of the DeWorm3 trials, evaluating the feasibility of interrupting STH transmission by comparing community-wide MDA to school-age-targeted deworming [28]. We estimate age-stratified community species-specific STH prevalence and describe individual, household and environmental factors associated with infection in this study population in Tamil Nadu.

## Methods

Reporting of this study has been verified in accordance with the Strengthening the Reporting of Observational Studies in Epidemiology (STROBE) checklist [29] (S1 STROBE Checklist).

### Ethical considerations

The DeWorm3 study was reviewed and approved by the Institutional Review Board at Christian Medical College, Vellore, India as well as the Institut de Recherche Clinique au Benin (IRCB) through the National Ethics Committee for Health Research, Ministry of Health in Benin, The London School of Hygiene and Tropical Medicine, The College of Medicine Research Ethics Committee in Malawi and the Human Subjects Division at the University of Washington. The trial is registered at ClinicalTrials.gov (NCT03014167). Prior to the initiation of the study in India, a technical review meeting was convened with government authorities at the national level and subsequently sensitization meetings were held with officials at the state, district and block levels. Meetings were held with local community leaders to explain the purpose of the study and procedures. Information sheets in Tamil, the local language, were provided to participants before each study activity. Written informed consent was sought from the household head for households' participation in the census. All LMC participants ≥18 years provided written informed consent while parental consent for participants <18 years was obtained along with verbal assent for children aged 7–11 years and written assent for children aged 12–17 years.

### Study setting

This study was carried out in two sub-sites in Tamil Nadu: the Timiri block in the Vellore Health Unit District (HUD) and selected villages in the Jawadhu Hills block of the Thiruvannamalai HUD (Fig 1). The last round of LF MDA was carried out in Vellore district in 2013 and Thiruvanamalai in 2015 [30]. The Timiri block is located in the plains of Vellore district and comprises four primary health centres (PHCs), each in turn divided into 25 health subcentres (HSCs). Each PHC serves a population of ~30,000 and each HSC ~5,000 respectively. This area has an average annual rainfall of 971 mm and has the following soil types; sandy and sandy loam 19%, red loam soil 20.8%, clay and clay loam 57.9% and black cotton soil 4.27% [31]. Although the most common occupation in this rural block is agriculture (20%), individuals also work in nearby industries as skilled and semi-skilled labourers. The Jawadhu Hills block comprises of three PHCs and is further subdivided into 13 HSCs with each PHC serving a mostly tribal population of ~20,000 and each HSC ~3,000 respectively. This block is located 762 metres above mean sea level in a reserve forest area. The tribal or indigenous people that populate this area are the Malayali and are classified as a scheduled tribe (disadvantaged communities or group of people listed in a schedule of the Indian constitution under article 342) by the government. The mean annual rainfall in Jawadhu Hills is 1100 mm, the mean maximum temperature is 36.6˚C and about 50% of the soil is red loamy clay and sandy soil. The main occupation is subsistence farming with more than 90% of the population involved in agricultural activities. Seasonal migration to nearby districts and states is common when residents work as semi-skilled labourers [32].

### Baseline census

The protocol and aims of the DeWorm3 trial have been previously published [28,33]. In the India site, 115 trained field workers conducted a baseline household census between October and December 2017 across 219 villages in Timiri and 154 villages in Jawadhu Hills. At each

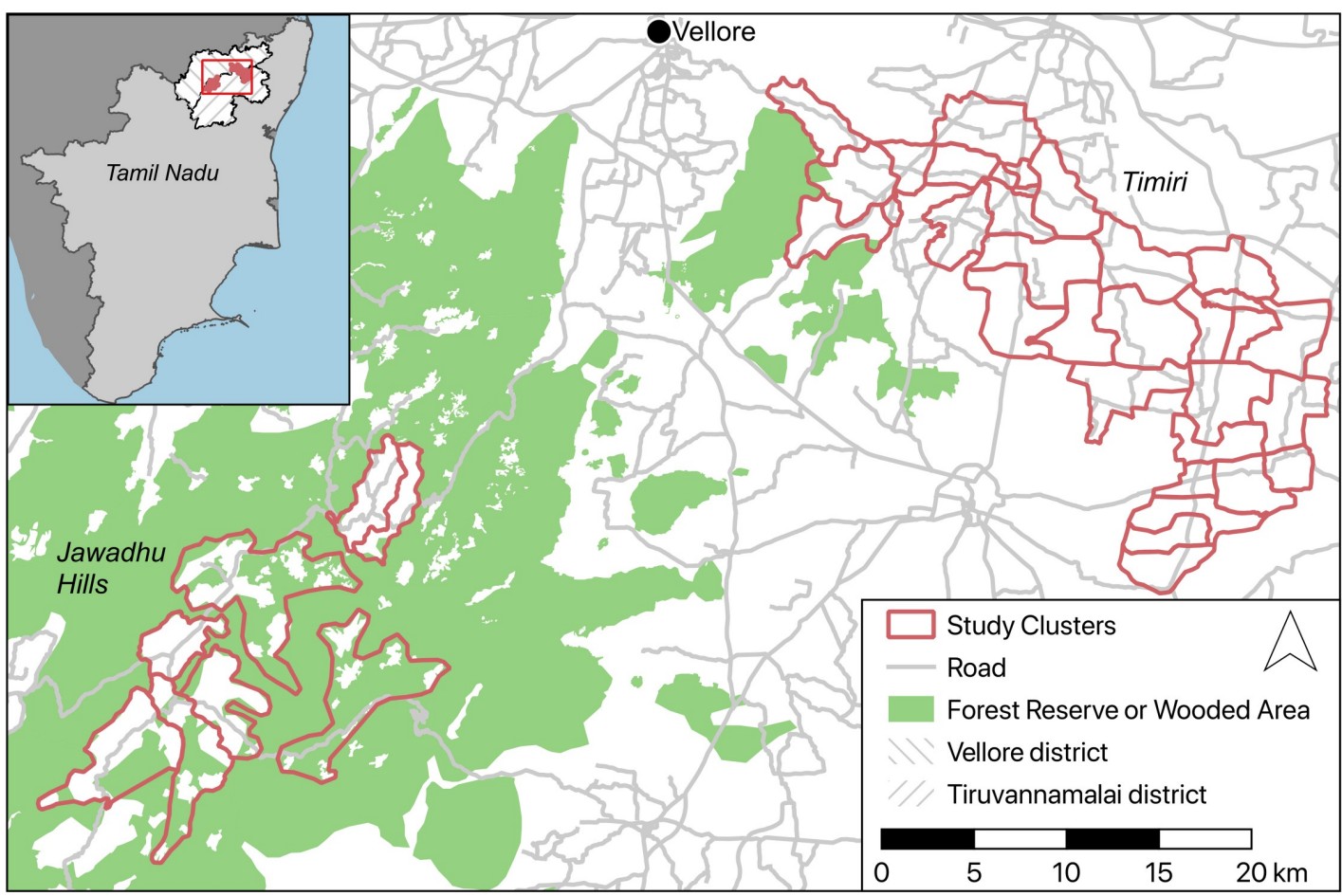

**Fig 1. Map of the Deworm3 India trial sub-sites at Timiri (32 clusters) and Jawadhu Hills (8 clusters) in Vellore and Thiruvanamalai districts\* of Tamil Nadu (inset).** \*District administrative boundaries obtained from gadm.org (https://gadm.org/license.html) and map data obtained from OpenStreetMap (https://www.openstreetmap.org/copyright).

household, following informed written consent provided by the household head or equivalent adult, household- and individual-level data were collected with a questionnaire programmed using SurveyCTO software (Dobility, Inc; Cambridge, MA and Ahmedabad, India) on an Android smartphone [34]. Demographic details such as age and sex were collected for all household members and were verified using state or central Government of India (GOI) issued identification cards (Aadhar card, Electoral Identity Card, Driving Licence or Birth Certificate). Household-level information including number of persons in the family, assets, sources of income and access to water and sanitation facilities were collected. Housing characteristics such as flooring, roof and walls were observed by field workers. All households were provided a study ID card that contained the household ID linked QR code sticker, name of the head of household and the address. Global Positioning System (GPS) coordinates were collected for all households censused, as well as for all structures at which no household members were found on three separate visits—these were classified as vacant or non-residential structures.

## Cluster demarcation

Following the census, all households were allocated to one of 40 study clusters. Contiguous cluster boundaries were confirmed on the basis of administrative and geographical

boundaries. Clusters were, on the whole, demarcated in line with HSCs, being divided along village boundaries where necessary, based on the requirement for clusters to comprise populations between 1,650 and 4,000. All villages within the study site boundaries were included. A total of 692 consented households from different health blocks that were located on the periphery of the cluster boundaries had been censused but were excluded during the demarcation process. A total of 32 clusters were defined in Timiri and 8 in Jawadhu Hills. In Jawadhu Hills, each HSC equated to a single cluster. While a majority of HSCs could be equated to single cluster in a Timiri, six HSCs were split into multiple clusters due to high population density.

## Survey design

Following the census and cluster demarcation, 6,000 individuals from the 40 clusters were recruited into a longitudinal monitoring cohort (LMC) to be followed up for five years. Age-stratified random sampling of 150 enumerated individuals with PSAC aged 1–4 years, SAC aged 5–14 years and adults aged 15 years and above in a ration of 1:1:3 was used to recruit 150 individuals in each cluster. All censused individuals above one year of age who were permanent residents, i.e. residing in the study area for more than six months and not planning to move out of the study area for the study duration, willing to provide informed consent (and assent where applicable) and provide samples, were eligible for recruitment. The survey was conducted between December 2017 to February 2018 in the local language spoken by all participants (Tamil) during which individual-level data including school enrolment, highest education level achieved, deworming history and shoe-wearing at the time of the survey were collected. Additionally, household water and sanitation facilities were observed, where possible, with indicators on construction materials and usage recorded. Data pertaining to household drinking water, sanitation facilities, and hygiene were collected according to the WHO UNICEF Joint Monitoring Program classification [35].

## Laboratory methods

A clean wide mouth container along with instructions on sample collection in Tamil, a wooden spatula and paper were provided to the participants. The containers had QR code stickers displaying the 9-digit participant ID, which was scanned upon receipt. The stool samples were transported on ice to the laboratory daily. All samples were read in duplicate by the Kato-Katz method by trained technicians who screened each slide (also labelled with the QR code sticker) for a minimum of six to eight minutes and within 30 minutes of preparation. The number of eggs in each slide was recorded on smartphones also using SurveyCTO software-based forms. The presence of other helminth ova and larvae was recorded but not quantified. With each batch, 10% of the slides were randomly checked by a supervisor for quality control. Presence of infection was determined if either one of the slides had at least one egg. Intensity of infection was calculated as the arithmetic mean of eggs per gram of faeces (EPG) by multiplying the eggs counted with a factor of 24 since the template used delivered 41.7 mg of stool. The WHO classification for intensity of infection for each of the species was used to categorise light, moderate and heavy intensity infections [36].

## Environmental data

Environmental and topographic data were explored as potential risk factors for infection [37]. Raster datasets on elevation and aridity at one $km^2$ resolution were obtained from the Consortium for Spatial Information [38]. Normalised Difference Vegetation Index (NDVI), Enhanced Vegetation Index (EVI), Middle Infrared (MIR) [39] and Land Surface Temperature (LST) [40] were produced by processing satellite images provided by the Moderate Resolution

Imaging Spectroradiometer (MODIS) instrument operating in the Terra spacecraft (NASA) at a resolution of 250m (NASA LP DAAC). Estimates of soil properties, such as sand fraction and soil acidity, were extracted from soilgrids.org at a resolution of 250m [41]. Environmental and topographic data were extracted using point-based extraction for each household using ArcGIS 10.3 (Environmental Systems Research Institute Inc. Redlands, CA, US).

## Statistical methods

Descriptive statistics of the baseline LMC characteristics were generated and prevalence was calculated in the three age categories (PSAC, SAC and adult). Age- and cluster-population-weighted estimates were calculated using the proportion of the censused population living in the cluster. Population density per $km^2$ was estimated through totalling the number of censused individuals falling within a one $km^2$ buffer placed around each household in ArcGIS. For households near study area boundaries, the number of censused individuals was divided by the buffer area falling within the boundary. Principal component analysis (PCA) in line with Filmer and Pritchett's widely used method [42] was used to arrive at a composite wealth index using various assets that were available to the households that included ownership of cooking fuel, electricity, radio, stove, DVD, television, computer, refrigerator, sofa set, mattress, solar lamp, ceiling fan, watch, mobile phone bicycle, motorcycle, autorickshaw, cart, car, livestock, house ownership, and housing materials. Cronbach's alpha assessed the dimensionality of the items included in the composite wealth index, and an item-rest correlation of 0.1 was set as a minimum threshold for including the item in the PCA. Variables with item-rest correlation less than 1 were removed and Cronbach's alpha value was computed again to see if all the variables were pointing to a similar direction with overall alpha set at 0.7. The wealth indices were divided into five SES quintiles, 1 being low and 5 being high. Household water source and sanitation facilities were categorised as improved and unimproved facilities for analyses according to JMP guidelines (JMP).

Univariable and multivariable mixed-effects multilevel logistic regression analysis was performed to build a model to assess the association between exposure factors and presence of infection, accounting for clustering at the household, village, and cluster level. Since STHs are highly aggregated and distributed in a negative binomial manner, the association between the intensity (EPG) of infection and associated factors was analysed using mixed-effects negative binomial regression method of the egg counts offsetting the actual quantity of stool used per sample and accounting for the clustering at all the levels. To fit the models, all significant (p<0.05) variables in the mixed-effects multilevel univariable analyses, were included in the multivariable model and a backwards stepwise approach was used to arrive at the most parsimonious models. Data with more than 500 missing values were excluded from multivariable analysis. Data management and analyses were performed using STATA version 16.0 (STATA Corporation, College Station, TX, USA).

## Results

The baseline census at the two sub-sites in India, Timiri and Jawadhu hills, enumerated 36,536 households comprising 140,932 individuals across an area of 477 $km^2$ between October and December 2017 (**Fig 2**). The demographic spread of the censused population was found to be skewed to the older ages, with 77.2% aged 15 years or above, the sex ratio was 1:1 and 30% of household heads reported having some secondary education or above. Only 34.6% of the households reported having access to improved sanitation, whereas 95.8% had access to an improved water source. Based on the census, the study area was then demarcated into 40 clusters. Final cluster sizes ranged from 494 to 1509 households and 2037 to 6002 individuals with

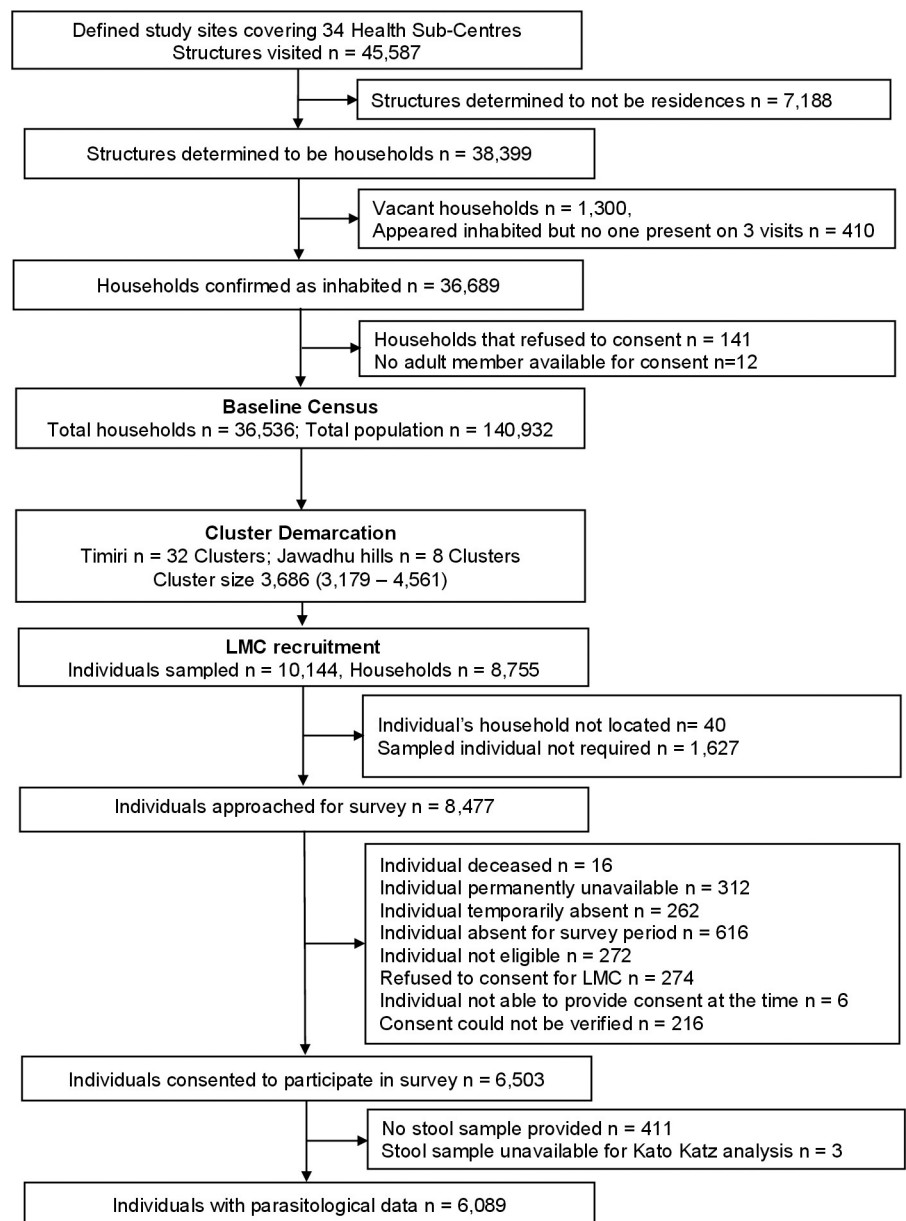

**Fig 2. Flow chart of activities during the baseline census and recruitment of the longitudinal monitoring cohort (October 2017- February 2018) Footnote: "Not required" refers to the number of individuals in the ranked, age-stratified, cluster-specific reserve/replacement lists generated who were not approached as the cohort sample size was achieved in the cluster.**

a mean of 989 (SD 273) and 3820 (SD 1072) households and individuals respectively. The median cluster area was 11.4 (interquartile range [IQR]: 7.5–15.7) km$^2$.

## Longitudinal monitoring cohort enrolment

From the age-stratified list of 10,144 individuals sampled for recruitment to the LMC, 8517 were approached between December 2017 and February 2018. Of these, 6998 (82.2%) were present and 6503 (92.9%) individuals consented to participate in the study. Among those who consented, 6370 (98.0%) completed the survey questionnaire and 6089 (93.6%) provided an

adequate stool sample for examination (**Fig 2**). Of the 6089 participants recruited from 5474 households in 368 villages, 1179 (19.4%), 1305 (21.4%), and 3605 (59.2%) were PSAC, SAC, and adults respectively. Recruitment of the target of 150 individuals was achieved in all the 40 clusters. When those who refused to participate in the LMC (n = 274) were compared to those who consented (n = 6503), the populations were broadly similar across characteristics, although the proportions of adults were higher in the group who refused (79.2% vs 59.9%), and in this group there was a higher proportion belonging to households in the high SES quintile (38.0% vs 22.6%), and where the head of the household had higher secondary or college education (18.3% vs 10.4%) (S1 Table). Among those who consented to participate but did not provide stool during the survey (n = 411), again, a higher proportion were adults (70.3%) than those who did provide a sample (59.2%).

## Characteristics of the LMC participants and their households at enrolment

The median age of the recruited participants in each category was 3.1 years (IQR: 2.2–4.0) among PSAC, 10.0 years (IQR: 7.3–12.6) among SAC and 40.0 years (IQR: 27.7–53.1) among adults. Participation among females (3252, 53.4%) was slightly higher compared to males (2836, 46.6%) (ratio 1.15). The LMC population was very closely representative of the census population, with the exception of age group, due to the age-stratified sampling. A third of participants came from households where the household head had no education. Among enrolled participants 3370 (55.4%) lived in houses made of man-made materials (improved) and 1510 (24.8%) lived in houses made of natural, non-durable materials (unimproved) with 10.4% and 9.4% living in houses made of mixed materials and government provided housing respectively. When flooring alone was categorized, 5370 (88.2%) had flooring of man-made materials as opposed to natural flooring. The majority of households used improved water source facilities (5824, 95.7%). However, 3968 or 65.2% of participants used unimproved sanitation facilities (including pit latrine without platform or open pit or open defaecation). Nearly half the individuals lived in households where livestock were owned (2745, 45.1%) and farming was carried out (2590, 44.5%).

## Prevalence and intensity of STH

The unweighted prevalence of any STH in the LMC at enrolment was 17% (95% CI: 16.0–17.9%) (**Table 1**). When weighted by age and cluster size the prevalence was 21.4% (95% CI: 20.4–22.4). Hookworm was the most common STH species detected with an unweighted prevalence of 16.6% (95% CI: 15.7–17.6) and weighted prevalence of 21% (95% CI: 20.0–22.2). Six individuals in the LMC had *Ascaris lumbricoides* and 17 had *Trichuris trichiura* infections, respectively, while two individuals with a dual infection were detected (*Ascaris* and hookworm). The mean intensity of hookworm infections was 634 EPG (SD 1493.6, median 198, IQR: 72–552) with EPG counts ranging from 12 to 18,756 EPG. Among the 6 individuals infected with *Ascaris*, the mean intensity was 1116 EPG (SD 1847.4, median 360, IQR: 216–1188) and ranged from 24 to 4848 EPG. Similarly, among the 17 individuals infected with *Trichuris*, the mean intensity was 209.6 EPG (SD 518.5, median 96, IQR 48–96) and ranged from 12 to 2196 EPG. Among the 1033 individuals who were positive for any STH, a very small proportion were found to have moderate to heavy intensity (MHI) infections for any STH species (n = 73, unweighted estimate 1.2%, 95% CI: 0.9–1.5%) and this was seen across all age categories (**Table 1**). Nearly all MHI were due to hookworm except for one *Trichuris* infection. Six other helminth species were also identified in 132 (2.2%) LMC participants during the survey and the more common species were *Enterobius vermicularis* (96, 1.6%) and *Hymenolepis nana* (27, 0.4%) (S2 Table).

**Table 1. Prevalence and intensity of soil transmitted helminths in the surveyed participants by age categories.**

| n = 6089 | Any STH | Hookworm | *Ascaris* | *Trichuris* |
|---|---|---|---|---|
| *Unweighted prevalence n (%)* | 1033 (17.0) | 1012 (16.6) | 6 (0.1) | 17 (0.3) |
| PSAC (1179) | 35 (3.0) | 31 (2.6) | 2 (0.2) | 2 (0.2) |
| SAC (1305) | 92 (7.0) | 87 (6.7) | 1 (0.1) | 5 (0.4) |
| Adult (3605) | 906 (25.1) | 894 (24.8) | 3 (0.1) | 10 (0.3) |
| *Intensity of infections among positives n (%)* | | | | |
| *PSAC (n = 35)* | | | | |
| Light-intensity | 32 (91.4) | 28 (90.3) | 2 (100) | 2 (100) |
| Moderate-intensity | 3 (8.6) | 3 (9.7) | 0 (0.0) | 0 (0.0) |
| Heavy-intensity | 0 (0.0) | 0 (0.0) | 0 (0.0) | 0 (0.0) |
| *SAC (n = 92)* | | | | |
| Light-intensity | 87 (94.6) | 83 (95.4) | 1 (100) | 4 (80.0) |
| Moderate-intensity | 4 (4.3) | 3 (3.4) | 0 (0.0) | 1 (20) |
| Heavy-intensity | 1 (1.1) | 1 (1.1) | 0 (0.0) | 0 (0.0) |
| *Adult (n = 906)* | | | | |
| Light-intensity | 841 (92.8) | 829 (92.7) | 3 (100) | 10 (100) |
| Moderate-intensity | 43 (4.7) | 43 (4.8) | 0 (0.0) | 0 (0.0) |
| Heavy-intensity | 22 (2.4) | 22 (2.5) | 0 (0.0) | 0 (0.0) |

## Individual and household characteristics associated with hookworm infection

As infections with *Ascaris* and *Trichuris* were very low, further analysis was carried out only for hookworm infections. The results of the univariable and multivariable mixed-effects logistic regression analysis for hookworm infection are presented in **Table 2**. In the univariable analysis, individual and household factors were associated with hookworm infection and nearly half of these variables remained significant in the multivariable mixed-effects logistic regression analysis. PSAC, SAC and adults had a hookworm prevalence of 2.6% (95% CI: 1.8–3.7), 6.7% (95% CI: 5.4–8.2), and 24.8% (95% CI: 23.4–26.2) respectively. In the multivariable regression, SAC (multivariable odds ratio [mOR] 3.8, 95% CI: 2.3–6.3) and adults (mOR 21.4, 95% CI: 12.3–37.2) were more likely to be infected (p<0.001) compared to PSAC (**Table 2, Fig 3**). Sex was not found to be associated with hookworm infection at the univariable or multivariable level. Among the other individual characteristics analysed, those who had a history of deworming in the past 12 months were less likely to be infected than those who did not (mOR 0.3, 95%CI: 0.2–0.5, p<0.001) but wearing shoes (based on observations during the survey) was not associated with reduced hookworm infection risk. After accounting for other variables, migratory status was no longer associated with hookworm infection and neither was livestock ownership or family size.

At the household level, a decreasing prevalence of hookworm was seen as education of the head of household increased, with a prevalence of 27.3% among those belonging to a household where the head had no education and 7.7% among those with household heads having higher secondary or college education. In the multivariable model the odds of infection were significantly lower among those with household heads having higher secondary or college education than in individuals from a household where the head had no education (p<0.001). Female literacy in the family also showed a similar correlation but was not included in the multivariable analysis. A decrease in odds of hookworm infection with increase in socioeconomic status of the household was also seen (mOR 0.3 per quintile, 95% CI: 0.2–0.5, p<0.001) (**Fig 4**). Although belonging to a household with flooring made of man-made materials was found

**Table 2.** Univariable and multivariable analysis of factors associated with hookworm infection.

| | Census (n = 140932) | LMC (n = 6089) | Hookworm infected (n = 1012) | Univariable | | Multivariable | |
|---|---|---|---|---|---|---|---|
| | n (%) | n (%) | n (%) | OR (95%CI) | p | mOR (95% CI) | p |
| **Individual factors** | | | | | | | |
| **Age** | | | | | | | |
| Infants (<1 year) | 1750 (1.2) | 0 (0) | 0 (0) | | | | |
| PSAC (1–4 years) | 8482 (6.0) | 1179 (19.4) | 31 (2.6) | 1 | <0.001 | 1 | **<0.001** |
| SAC (5–14 years) | 21839 (15.5) | 1305 (21.4) | 87 (6.7) | 3.3 (2.0, 5.4) | | 3.8 (2.3,6.3) | |
| Adult (15+ years) | 108861 (77.2) | 3605 (59.2) | 894 (24.8) | 27.2 (15.3, 48.3) | | 21.4 (12.3, 37.2) | |
| **Sex*** | | | | | | | |
| Male | 70295 (49.9) | 2836 (46.6) | 460 (16.2) | 1 | 0.215 | | |
| Female | 70620 (50.1) | 3252 (53.4) | 552 (17.0) | 1.1 (0.9, 1.3) | | | |
| **Lived there most of past 6 months** | | | | | | | |
| Yes | 137144 (97.3) | 6041 (99.2) | 1006 (16.7) | 1 | 0.037 | | |
| No | 3788 (2.7) | 48 (0.8) | 6 (12.5) | 0.3 (0.1, 0.9) | | | |
| **Slept there last night** | | | | | | | |
| Yes | 133153 (94.5) | 5975 (98.1) | 992 (16.6) | 1 | 0.225 | | |
| No | 7779 (5.5) | 114 (1.9) | 20 (17.5) | 0.7 (0.4, 1.3) | | | |
| **Wearing shoes during survey†** | | | | | | | |
| No | - | 3273 (54.8) | 609 (18.6) | 1 | 0.899 | | |
| Yes | - | 2698 (45.2) | 390 (14.5) | 1.0 (0.8, 1.2) | | | |
| **Dewormed in last 12 months†** | | | | | | | |
| No | - | 4963 (83.1) | 942 (19.0) | 1 | <0.001 | 1 | **<0.001** |
| Yes | - | 1008 (16.9) | 57 (5.7) | 0.2 (0.1, 0.2) | | 0.3 (0.2,0.5) | |
| **Household factors** | | | | | | | |
| **Population density ‡** | | | | | | | |
| <50 | 6750 (4.8) | 262 (4.3) | 79 (30.1) | 1 | 0.416 | | |
| 50–249 | 70424 (50.0) | 3091 (50.8) | 648 (21.0) | 0.9 (0.6, 1.3) | | | |
| 250–999 | 55013 (39.1) | 2386 (39.2) | 268 (11.2) | 0.9 (0.6, 1.4) | | | |
| > = 1000 | 8956 (6.1) | 344 (5.7) | 16 (4.7) | 0.5 (0.2, 1.2) | | | |
| **Family size** | | | | | | | |
| < = 4 members | 71670 (50.9) | 2899 (47.6) | 556 (19.5) | 1 | <0.001 | | |
| > = 5 members | 69262 (49.2) | 3190 (52.4) | 446 (14.0) | 0.6 (0.5, 0.7) | | | |
| **Farming household§** | | | | | | | |
| No | 75470 (56.1) | 3235 (55.5) | 571 (17.7) | 1 | 0.082 | | |
| Yes | 59027 (43.9) | 2590 (44.5) | 414 (16.0) | 1.2 (1.0, 1.4) | | | |
| **Livestock possession** | | | | | | | |
| No | 78179 (55.5) | 3344 (54.9) | 446 (13.3) | 1 | 0.197 | | |
| Yes | 62753 (44.5) | 2745 (45.1) | 566 (20.6) | 1.1 (0.9, 1.3) | | | |
| **Highest education level of any female family member¶** | | | | | | | |
| No education | 38426 (27.5) | 1509 (25.0) | 463 (30.7) | 1 | <0.001 | | |
| Some primary | 18894 (13.5) | 812 (13.5) | 143 (17.6) | 0.7 (0.6, 1.0) | | | |
| Some middle | 24089 (17.3) | 1091 (18.1) | 139 (12.7) | 0.5 (0.3, 0.6) | | | |
| Some secondary | 27731 (19.9) | 1318 (21.9) | 141 (10.7) | 0.4 (0.3, 0.5) | | | |
| Some higher secondary / college | 30451 (21.8) | 1301 (21.6) | 112 (8.6) | 0.3 (0.2, 0.4) | | | |

*(Continued)*

**Table 2.** (Continued)

| | Census (n = 140932) | LMC (n = 6089) | Hookworm infected (n = 1012) | Univariable | | Multivariable | |
|---|---|---|---|---|---|---|---|
| | n (%) | n (%) | n (%) | OR (95%CI) | p | mOR (95% CI) | p |
| **Highest education level of household head#** | | | | | | | |
| No education | 46198 (33.0) | 1835 (30.4) | 500 (27.3) | 1 | <0.001 | 1 | **<0.001** |
| Some primary | 25705 (18.4) | 1121 (18.6) | 167 (14.9) | 0.6 (0.5, 0.8) | | 0.7 (0.6, 1.0) | |
| Some middle | 26124 (18.7) | 1186 (19.6) | 158 (13.3) | 0.5 (0.4, 0.7) | | 0.7 (0.5, 0.9) | |
| Some secondary | 27900 (19.9) | 1267 (21.0) | 129 (10.2) | 0.4 (0.3, 0.5) | | 0.6 (0.5, 0.8) | |
| Some higher secondary / college | 14112 (10.1) | 634 (10.5) | 49 (7.7) | 0.3 (0.2, 0.4) | | 0.4 (0.3, 0.6) | |
| **Household floor**\*\* | | | | | | | |
| Natural materials | 16334 (11.6) | 708 (11.7) | 220 (31.1) | 1 | 0.004 | | |
| Manmade materials | 124437 (88.4) | 5370 (88.4) | 791 (14.7) | 0.7 (0.5, 0.9) | | | |
| **SES (1 = Low, 5 = High)** | | | | | | | |
| 1 | 23884 (16.9) | 937 (15.4) | 364 (38.9) | 1 | <0.001 | 1 | **<0.001** |
| 2 | 26674 (18.9) | 1100 (18.1) | 228 (20.7) | 0.5 (0.4, 0.7) | | 0.6 (0.4, 0.8) | |
| 3 | 27545 (19.5) | 1251 (20.6) | 165 (13.2) | 0.4 (0.3, 0.6) | | 0.6 (0.4, 0.8) | |
| 4 | 30689 (21.8) | 1424 (23.4) | 147 (10.3) | 0.3 (0.2, 0.5) | | 0.4 (0.3, 0.6) | |
| 5 | 32140 (22.8) | 1377 (22.6) | 108 (7.8) | 0.2 (0.2, 0.3) | | 0.3 (0.2, 0.5) | |
| **Water source** | | | | | | | |
| Unimproved facilities | 5937 (4.2) | 265 (4.4) | 116 (43.8) | 1 | 0.213 | | |
| Improved facilities | 134995 (95.8) | 5824 (95.7) | 896 (15.4) | 0.8 (0.5, 1.2) | | | |
| **Sanitation facilities** | | | | | | | |
| Unimproved facilities | 92216 (65.4) | 3968 (65.2) | 821 (20.7) | 1 | <0.001 | | |
| Improved facilities | 48716 (34.6) | 2121 (34.8) | 191 (9.0) | 0.6 (0.5, 0.8) | | | |
| **Handwashing facility**§§ | | | | | | | |
| No | - | 3687 (66.3) | 734 (19.9) | 1 | <0.001 | | |
| Yes | - | 1873 (33.7) | 195 (10.4) | 0.6 (0.5, 0.7) | | | |
| **Environmental factors** | | | | | | | |
| **Mean normalised difference vegetation index (NDVI)‡** | | | | | | | |
| First tertile | 47073 (33.4) | 1959 (32.2) | 268 (13.7) | 1 | 0.031 | 1 | **0.007** |
| Second tertile | 46850 (33.3) | 2059 (33.9) | 332 (16.1) | 1.4 (1.1, 1.7) | | 1.5 (1.2, 2.0) | |
| Third tertile | 46860 (33.3) | 2065 (34.0) | 411 (19.9) | 1.4 (1.1, 1.8) | | 1.4 (1.1, 1.9) | |
| **Mean of middle infrared (MIR)‡** | | | | | | | |
| First tertile | 47136 (33.5) | 2056 (33.8) | 424 (20.6) | 1 | 0.175 | | |
| Second tertile | 46761 (33.2) | 2071 (34.1) | 328 (15.8) | 0.8 (0.6, 1.1) | | | |
| Third tertile | 46886 (33.3) | 1956 (32.2) | 259 (13.2) | 0.8 (0.6, 1.0) | | | |
| **Elevation‡** | | | | | | | |
| First tertile | 47036 (33.4) | 2053 (33.8) | 245 (11.9) | 1 | <0.001 | 1 | **<0.001** |
| Second tertile | 47455 (33.7) | 2014 (33.1) | 151 (7.5) | 1.0 (0.6, 1.6) | | 0.9 (0.6, 1.5) | |
| Third tertile | 46292 (32.9) | 2016 (33.1) | 615 (30.5) | 4.2 (2.4, 7.2) | | 3.9 (2.3, 6.8) | |
| **Sand fraction‡** | | | | | | | |
| First tertile | 50457 (35.8) | 2436 (40.1) | 313 (12.9) | 1 | 0.177 | | |
| Second tertile | 50588 (35.9) | 1643 (27.0) | 193 (11.8) | 0.8 (0.7, 1.1) | | | |
| Third tertile | 39738 (28.2) | 2004 (32.9) | 505 (25.2) | 1.1 (0.8, 1.5) | | | |
| **Soil acidity (pH KCL)‡** | | | | | | | |
| First tertile | 66905 (47.5) | 2656 (43.7) | 605 (22.8) | 1 | 0.289 | | |
| Second tertile | 28233 (20.1) | 1596 (26.2) | 167 (10.5) | 0.8 (0.6, 1.1) | | | |

(*Continued*)

**Table 2.** (Continued)

| | Census (n = 140932) | LMC (n = 6089) | Hookworm infected (n = 1012) | Univariable | | Multivariable | |
|---|---|---|---|---|---|---|---|
| | n (%) | n (%) | n (%) | OR (95%CI) | p | mOR (95% CI) | p |
| Third tertile | 45645 (32.4) | 1831 (30.1) | 239 (13.1) | 0.8 (0.5, 1.2) | | | |

\*1 missing value

†118 missing values

‡ 6 missing values

§ 264 missing values

¶ 58 missing values and not included in the multivariable model as education of the head of the household was included

# 46 missing values

\*\*11 missing values

§§529 missing values, not included in the multivariable model due to high number of missing values

Natural flooring includes earth, dung, palm or bamboo and stone; Man-made flooring includes wood, brick, vinyl or asphalt strips, tiles, cement, carpet and polished stone like marble or granite

Improved water source refers to limited and basic facilities, Unimproved refers to surface water and unimproved facilities

Improved sanitation refers to limited and basic facilities, Unimproved refers to open defaecation and unimproved facilities

Acronyms: LMC- Longitudinal Monitoring Cohort, OR–odds ratio, mOR–multivariable OR, 95% CI—95% confidence interval

to be associated with decreased odds initially (OR 0.7, 95%CI: 0.5–0.9, p = 0.004), after accounting for other variables, this did not remain significant in the multivariable analysis. When household WASH factors were analysed in the univariable analysis, those belonging to households with improved sanitation and having handwashing facilities had a reduced odds of infection, (OR 0.6, 95% CI: 0.5–0.8, p<0.001) and (OR 0.6, 95% CI: 0.5–0.7 p<0.001) respectively. Sanitation did not remain significant in the multivariable analysis and handwashing was not analysed further due to missing data.

### Factors associated with intensity of hookworm infection

The mean (SD) EPG in PSAC, SAC and adults was 538.8 (847.7), 389.7 (746.3), 661.1 (1562.3) respectively with a trend of increased EPG in adults seen in both sexes (Fig 3). In the multivariable analysis (Table 3), an increased infection intensity ratio (IIR) was seen for both SAC (multivariable IIR [mIIR] 9.2, 95% CI: 4.3–19.7) and adults (mIIR 332.5, 95% CI: 166.4–664.5, p<0.001) compared to PSAC. Although female sex appeared associated with higher intensity infections, after accounting for other associated factors, this relationship was no longer significant. Those who had a history of deworming in the past 12 months had a lower EPG (mIIR 0.1, 95% CI: 0.1–0.2, p<0.001). No other factors at the individual level were associated with intensity of infection.

At the household level, decreasing intensity of hookworm as education of the head of household increased was seen (mIIR 0.2, 95%CI: 0.1–0.5, p<0.001). Female education also showed a similar correlation with intensity of infection. The association with SES was also similar to that seen with presence of infection (Fig 4) with a decreasing intensity of hookworm infection with increase in socioeconomic status of the household (mIIR 0.2, 95% CI: 0.1–0.4, p<0.001). When household WASH factors were analysed, those belonging to households with improved sanitation had a lower infection intensity than those residing in households with unimproved facilities (mIIR 0.6, 95%CI: 0.4–0.9). Belonging to households with a handwashing facility was also associated with lower intensity compared to households that did not have facilities in the univariable analysis (IIR 0.3, 95% CI: 0.2–0.5).

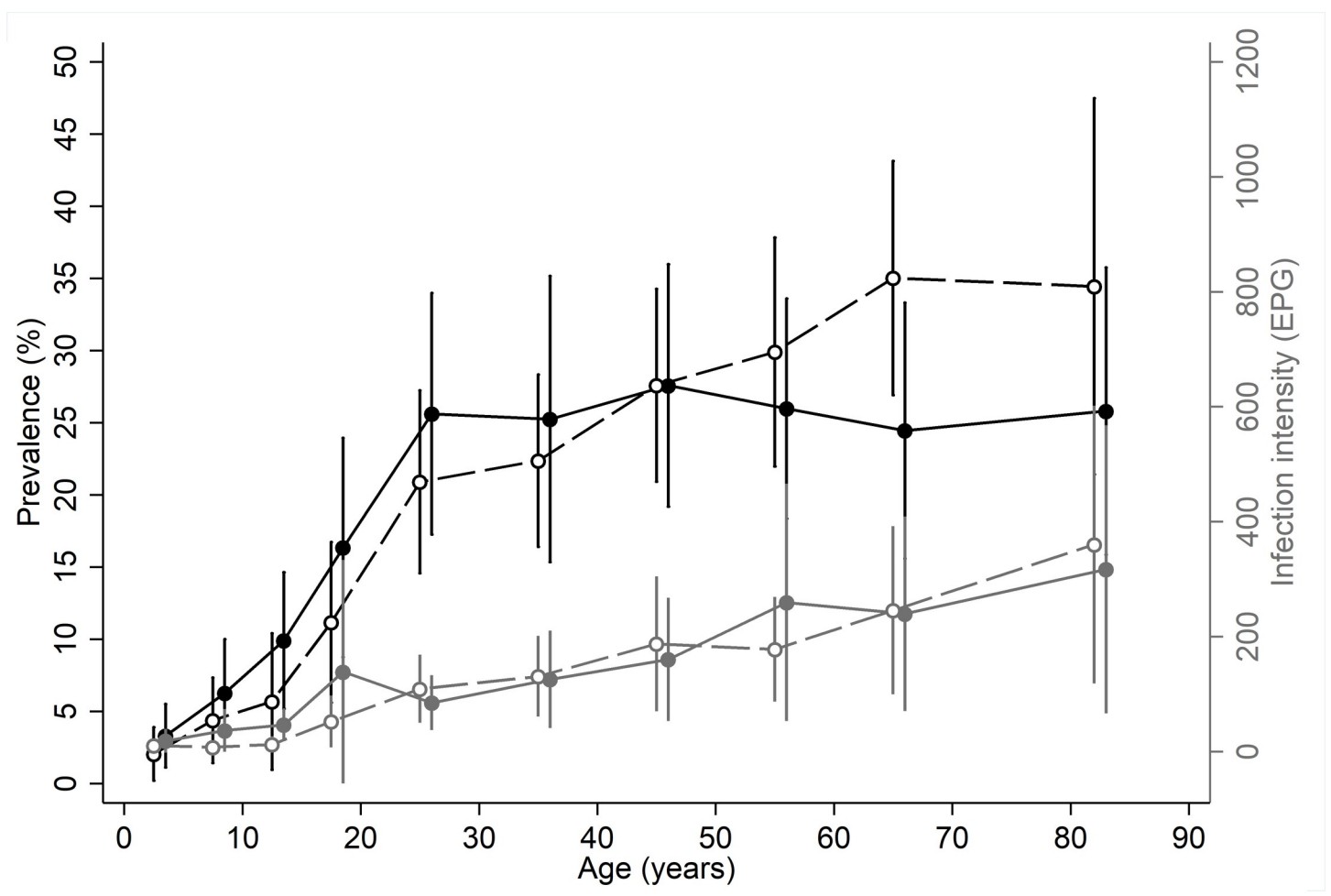

**Fig 3.** Age-infection profiles of hookworm among surveyed participants at enrolment—prevalence (black lines) and intensity (grey lines) of hookworm infection for males (solid line and circles) and females (dashed lines and empty circles).

### Environmental risk factors associated with presence and intensity of hookworm infection

Assessment of environmental factors in the multivariable analysis indicated that higher vegetation coverage (Normalized Difference Vegetation Index or NDVI) (mOR 1.4, 95% CI: 1.1–1.9) and higher elevation (mOR 3.9; 95% CI: 2.3–6.8) were associated with the increased odds of hookworm infection. Upon assessing the environmental factors associated with intensity of infection the same parameters were associated with increased egg counts (mIIR for NDVI 2.4, 95%CI: 1.4–3.9, p<0.001 and for elevation, mIIR 14.1, 95%CI: 6.5–30.7, p<0.001). Among environmental parameters, the aridity index was not included in the analysis as all households in the study site were in the same sub-humid category (range among households 0.54–0.62). Enhanced vegetation index (EVI) and land surface temperature (LST) were also excluded as they were highly correlated with NDVI and elevation respectively.

## Discussion

The results of this parasitological survey conducted in an age-stratified cohort of 6089 individuals nested within the censused Deworm3 trial population of 140,932 individuals in southern

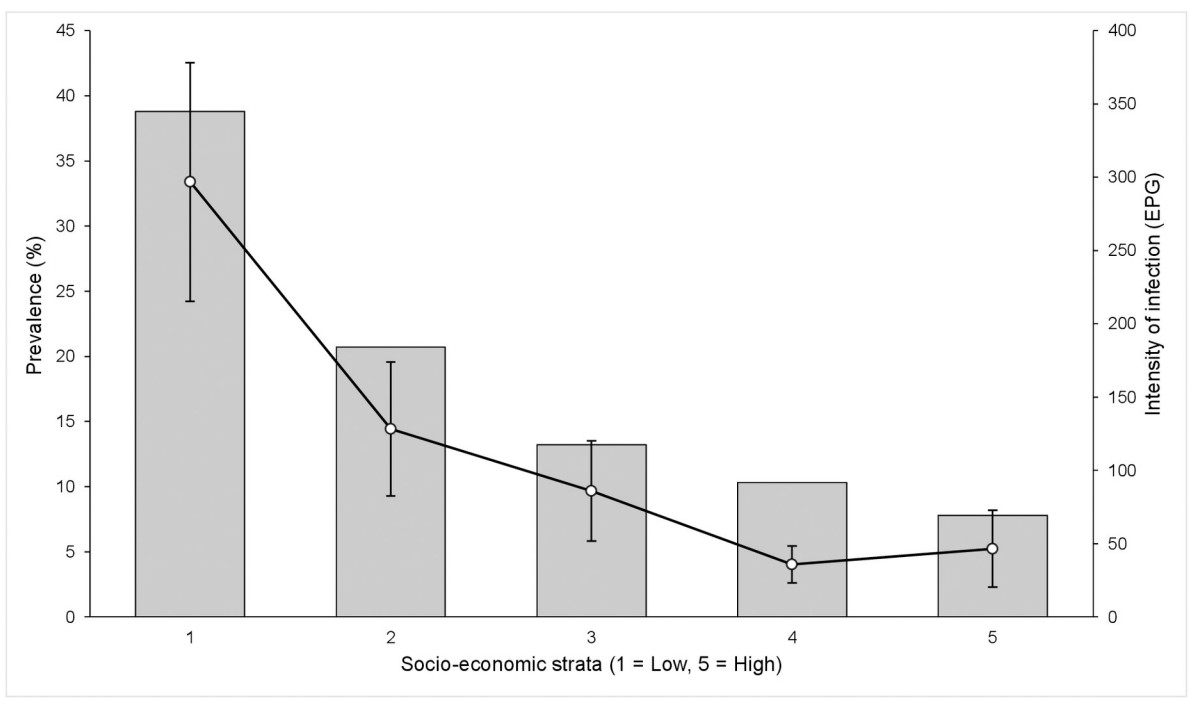

**Fig 4.** Prevalence and intensity of hookworm among surveyed participants stratified by categories of socio-economic status—prevalence (bars) and intensity (lines).

India showed that hookworm was the most common STH infection in this region. The prevalence of hookworm was high and consequently remains a significant public health problem. Our study site is unique as it incorporates two subsites—a rural plain area of 32 clusters in Timiri and a difficult-to-reach, hilly area of 8 clusters in Jawadhu Hills with a mostly tribal population. The censused population comprehensively described the communities and age demographic profile that is typical of the region and the LMC surveyed was representative of the population enumerated. Our study indicated that a range of individual, household and environmental factors in these communities including age, SES, education, sanitation and vegetation influence both the odds of infection and the intensity of infection.

As reported in the pooled analysis [43], the age-weighted prevalence of STH in the India site was substantially higher (21.4%) than both the Malawi and Benin study sites, with the vast majority of infections attributable to hookworm. Only a small proportion of infections were of MHI (1.5%). Due to the low number of *Ascaris* and *Trichuris* infections detected, further analysis was limited to factors associated with hookworm prevalence and intensity in this population. While no previous data are available from the Timiri area, previous studies at Jawadhu Hills have shown a high prevalence of hookworm—38% in 2011–12 and 18.5% in 2013–14—despite multiple rounds of treatment with albendazole in the district as part of the LF control program [44,45]. Data collected from the neighbouring district, Villupuram, in Tamil Nadu in 2000, prior to the commencement of the LF programme involving combined administration of Albendazole, found an STH prevalence of 60% among children aged 9–10 years, with particularly high *Ascaris* prevalence. A 70% decrease in STH prevalence after three rounds of MDA with DEC and Albendazole was recorded among SAC [46]. These reductions in prevalence over time and the very low proportion of MHI infections observed in this study would suggest that, despite the ongoing transmission of hookworm, the LF programme and the school-based

**Table 3. Univariable and multivariable analysis of factors associated with intensity of hookworm infection.**

| | EPG (N = 1012)* | Univariable | | Multivariable | |
|---|---|---|---|---|---|
| | Arithmetic mean (SD) | IIR (95%CI) | p | mIIR (95%CI) | p |
| **Individual factors** | | | | | |
| **Age** | | | | | |
| PSAC (1–4 years) | 538.8 (847.7) | 1 | <0.001 | 1 | **<0.001** |
| SAC (5–14 years) | 389.7 (746.3) | 7.4 (3.6, 15.5) | | 9.2 (4.3, 19.7) | |
| Adult (15+ years) | 661.1 (1562.3) | 525.2 (259.3, 1063.8) | | 332.5 (166.4, 664.5) | |
| **Sex†** | | | | | |
| Male | 664.7 (1682.5) | 1 | 0.017 | | |
| Female | 608.4 (1316.6) | 1.5 (1.1, 2.0) | | | |
| **Lived there most of past 6 months** | | | | | |
| Yes | 530.0 (978.0) | 1 | 0.058 | | |
| No | 634.6 (1496.4) | 0.2 (0.0, 1.1) | | | |
| **Slept there the previous night** | | | | | |
| Yes | 480.6 (742.9) | 1 | 0.213 | | |
| No | 637.1 (1504.9) | 0.5 (0.1, 1.5) | | | |
| **Wearing shoes during survey‡** | | | | | |
| No | 638.9 (1509.1) | 1 | 0.304 | | |
| Yes | 618.9 (1473.6) | 0.8 (0.5, 1.2) | | | |
| **Dewormed in last 12 months‡** | | | | | |
| No | 636.8 (1503.3) | 1 | <0.001 | 1 | **<0.001** |
| Yes | 536.4 (1352.1) | 0.0 (0.0, 0.0) | | 0.1 (0.1, 0.2) | |
| **Household factors** | | | | | |
| **Population density§** | | | | | |
| <50 | 593.5 (792.2) | 1 | 0.015 | | |
| 50–249 | 659.2 (1632.9) | 1.0 (0.5, 2.3) | | | |
| 250–999 | 610.9 (1336.7) | 0.8 (0.3, 1.9) | | | |
| > = 1000 | 219.8 (213.8) | 0.2 (0.1, 0.7) | | | |
| **Family size** | | | | | |
| < = 4 members | 652.9 (1374.5) | 1 | <0.001 | | |
| > = 5 members | 610.1 (1633.5) | 0.4 (0.3, 0.6) | | | |
| **Farming household¶** | | | | | |
| No | 722.4 (1691.4) | 1 | 0.483 | | |
| Yes | 534.1 (1210.2) | 1.1 (0.8, 1.6) | | | |
| **Livestock possession** | | | | | |
| No | 627.6 (1214.9) | 1 | 0.178 | | |
| Yes | 639.0 (1682.0) | 1.3 (0.9, 1.7) | | | |
| **Highest education level of any female family member#** | | | | | |
| No education | 718.6 (1682.3) | 1 | <0.001 | | |
| Some primary | 460.0 (962.4) | 0.5 (0.3, 0.9) | | | |
| Some middle | 652.6 (1361.2) | 0.4 (0.2, 0.6) | | | |
| Some secondary | 624.9 (1736.1) | 0.2 (0.1, 0.4) | | | |
| Some higher secondary / college | 475.2 (1012.7) | 0.1 (0.1, 0.2) | | | |
| **Highest education level of household head**** | | | | | |
| No education | 670.8 (1563.8) | 1 | <0.001 | 1 | **<0.001** |
| Some primary | 777.1 (1897.2) | 0.5 (0.3, 0.8) | | 0.6 (0.4, 1.0) | |
| Some middle | 592.3 (1132.1) | 0.4 (0.2, 0.6) | | 0.5 (0.3, 0.8) | |
| Some secondary | 385.5 (1111.8) | 0.2 (0.1, 0.3) | | 0.4 (0.2, 0.6) | |

(*Continued*)

**Table 3.** (Continued)

| | EPG (N = 1012)* | Univariable | | Multivariable | |
|---|---|---|---|---|---|
| | Arithmetic mean (SD) | IIR (95%CI) | p | mIIR (95%CI) | p |
| Some higher secondary / college | 640.4 (1150.9) | 0.1 (0.1, 0.2) | | 0.2 (0.1, 0.5) | |
| **Household floor‡‡** | | | | | |
| Natural materials | 669.1 (1588.3) | 1 | 0.013 | | |
| Manmade materials | 624.6 (1468.0) | 0.5 (0.3, 0.9) | | | |
| **SES (1 = Low, 5 = High)** | | | | | |
| 1 | 764.0 (1786.7) | 1 | <0.001 | 1 | **<0.001** |
| 2 | 618.6 (1555.6) | 0.5 (0.3, 0.9) | | 0.5 (0.3, 0.8) | |
| 3 | 651.7 (1327.8) | 0.4 (0.2, 0.7) | | 0.5 (0.3, 0.9) | |
| 4 | 346.2 (672.9) | 0.2 (0.1, 0.3) | | 0.3 (0.1, 0.5) | |
| 5 | 593.1 (1272.9) | 0.1 (0.1, 0.2) | | 0.2 (0.1, 0.4) | |
| **Water source‡‡** | | | | | |
| Unimproved facilities | 805.2 (2128.6) | 1 | 0.877 | | |
| Improved facilities | 611.8 (1390.5) | 1.1 (0.4, 2.6) | | | |
| **Sanitation facilities§§** | | | | | |
| Unimproved facilities | 655.0 (1564.5) | 1 | <0.001 | 1 | **0.016** |
| Improved facilities | 543.6 (1138.5) | 0.4 (0.3, 0.6) | | 0.6 (0.4, 0.9) | |
| **Handwashing facility¶¶** | | | | | |
| No | 683.2 (1633.7) | 1 | <0.001 | | |
| Yes | 465.7 (858.6) | 0.3 (0.2, 0.5) | | | |
| **Environmental factors** | | | | | |
| **Mean normalised difference vegetation index (NDVI)§** | | | | | |
| First tertile | 511.8 (1069.3) | 1 | 0.018 | 1 | **<0.001** |
| Second tertile | 553.4 (1392.1) | 1.8 (1.2, 2.9) | | 2.4 (1.5, 3.8) | |
| Third tertile | 779.6 (1776.3) | 1.8 (1.1, 2.9) | | 2.4 (1.4, 3.9) | |
| **Mean of middle infrared (MIR)§** | | | | | |
| First tertile | 786.0 (1878.8) | 1 | 0.515 | | |
| Second tertile | 574.5 (1190.3) | 0.8 (0.5, 1.4) | | | |
| Third tertile | 461.8 (1042.0) | 0.7 (0.4, 1.3) | | | |
| **Elevation (N = 6,083)§** | | | | | |
| First tertile | 476.4 (941.3) | 1 | <0.001 | 1 | **<0.001** |
| Second tertile | 458.2 (1193.4) | 0.9 (0.4, 1.8) | | 1.0 (0.5, 2.0) | |
| Third tertile | 740.5 (1715.8) | 8.1 (3.4, 19.3) | | 14.1 (6.5, 30.7) | |
| **Sand fraction (N = 6,083)§** | | | | | |
| First tertile | 610.1 (1433.8) | 1 | 0.348 | | |
| Second tertile | 639.9 (1575.4) | 1.0 (0.7, 1.6) | | | |
| Third tertile | 647.3 (1502.0) | 1.5 (0.8, 2.6) | | | |
| **Soil acidity (pH KCL)§** | | | | | |
| First tertile | 680.7 (1679.0) | 1 | 0.169 | | |
| Second tertile | 576.6 (1300.1) | 0.6 (0.4, 1.1) | | | |

(*Continued*)

**Table 3.** (Continued)

|  | EPG (N = 1012)* | Univariable |  | Multivariable |  |
|---|---|---|---|---|---|
|  | Arithmetic mean (SD) | IIR (95%CI) | p | mIIR (95%CI) | p |
| Third tertile | 557.2 (1062.5) | 0.6 (0.3, 1.2) |  |  |  |

*EPG is presented for the infected individuals only, but the negative binomial models include all individuals regardless of infection status

† 1 missing value

‡ 118 missing values

§ 6 missing values

¶ 264 missing values

# 58 missing values and data not included in the multivariable model as education of the head of the household was included

** 46 missing values

†† 11 missing values

¶¶529 missing values, not included in the multivariable model due to high number of missing values

Natural flooring refers to earth, dung, palm or bamboo and stone

Man-made flooring includes wood, brick, vinyl or asphalt strips, tiles, cement, carpet and polished stone like marble or granite

Improved water source refers to limited and basic facilities, Unimproved refers to surface water and unimproved facilities

Improved sanitation refers to limited and basic facilities, Unimproved refers to open defaecation and unimproved facilities

Acronyms: EPG—egg per gram, IIR–Infection intensity ratio, mIIR–multivariable IIR, 95% CI—95% confidence interval

deworming programmes have been effective in reducing heavy intensity hookworm infections and possibly the prevalence of *Ascaris* in SAC.

The analyses highlighted several important correlates of hookworm infection in this setting. One of the most prominent of these was age, which was associated both with increased odds of infection as well as higher intensity of infection. This age intensity profile associated with hookworm has been described previously in several studies [37,47,48]. In our analyses, sex was not associated with hookworm infection. While adult males are sometimes found to be at higher risk for hookworm infection [49], previous studies conducted in this region have similarly found no association between sex and hookworm infection [44,50]. Although not significant, the prevalence of infection by sex was similar until the 4th decade after which women had a higher prevalence than men. An increase in intensity of infection but not prevalence in older women has been noted in previous studies [47,51]. As expected, a history of deworming was associated with substantially (70%) reduced infection prevalence and intensity. This result is in line with the majority of community surveys conducted within the context of an ongoing school deworming programme [37,50] and highlights the successes of the strategy in the target group, which in the India NDD extends further than in many endemic countries (1 to 19 year olds included).

A higher level of education of the head of household and high SES were both independently strongly associated with decreased odds of infection as well as decreased intensity of infection. These findings are similar to many other community STH studies [47,52]. SES is closely related to several other household-level factors measured in the survey, one such example is flooring. Although a manmade floor was found to be associated with lower odds of infection in univariate analysis, floor type was not significantly associated after accounting for SES. This is likely because flooring was highly correlated with other housing construction variables included in the SES composite variable.

With India currently implementing the world's largest sanitation programme, the Swachh Bharat Mission, there has been an unprecedented scale of toilet building but functionality and uptake remain challenges [20]. In a study in rural Odisha, India increased community sanitation coverage did not reduce diarrheal disease or acute respiratory infections but a reduction

in prevalence of helminth infections was seen along with a reduction in stunting in children under 5 [53]. In our study, although unimproved sanitation (including open defecation) was not associated with an increased odds of hookworm infection, residing in a household with access to improved sanitation was associated with decreased intensity of infection. Soil samples from a smaller proportion of households in the study site have been collected to quantify environmental STH contamination. These results will be presented in a future paper and may be useful in elucidating the relationships between sanitation access, peri-domestic risk and intensity of infection. Access to a facility to wash hands with soap and water in the household was also associated with decreased risk of infection as well as lower intensity of infection but was not included in the final multivariable model as data were not available for 590 households. In a meta-analysis of studies that applied JMP definitions to categorize WASH facilities, both access to sanitation as well as access to water and hygiene facilities were associated with reduced odds of infection [54]. While the effect of WASH interventions are not easily evaluated especially in the context of other interventions [55], the importance of integrating comprehensive behavioural and structural WASH interventions and access to potentially sustain the gains made from deworming in the longer term has been highlighted in a recent modelling study [56].

Environmental factors that affect temperature, soil moisture and atmospheric humidity influence the rate of survival and development of hookworm larvae thereby affecting transmission [57]. In this study, both increased vegetation (NDVI) and elevation were associated with increased odds of infection as well as an increase in intensity of infection. In another study using remotely sensed data at a fine resolution in Jawadhu Hills, topographical parameters of elevation and slope were negatively and positively associated with hookworm infections at the village level [58]. Riess *et al.* have shown that ecological variables are associated with hookworm infection but have differing effects within a geographical region, are scale-dependent and urge caution against prediction at smaller scales using large-scale data [52]. Moreover, the effect of elevation in the current study is likely to be associated with the topographical differences between the Timiri and Jawadhu subsites and needs to be explored further using a finer resolution approach.

A robust study design was used to assess the burden of STH among community members in the study site. However, these results presented here are based on a parasitological survey conducted using the Kato Katz technique on a single stool sample, which has been shown to be sub-optimal in estimating prevalence, especially in low intensity settings [59,60]. This limitation will be addressed by future analyses on these samples using field-validated high throughput species specific qPCR [61]. Tribal communities have previously been shown to have higher STH transmission than plains populations, especially urban populations [62]. Further analyses by sub-site and using spatial analyses would be useful to tease out these additional correlates of risk in these different settings and highlight heterogeneity in infection risk. These analyses are not possible at this stage of the trial due to blinding restrictions.

The findings presented here highlight that despite several years of community-based deworming through the LF programme and multiple rounds of school-based deworming, community transmission of hookworm is still persisting in both rural and tribal areas of Tamil Nadu, especially in adults. This study provides important, robust data that will be useful to the research community as well as the Ministry of Health and Family Welfare in planning future potential expansion of the deworming program with synergy across other initiatives including the anemia free India targeting WRA, MDA for lymphatic filariasis in endemic districts and the recently launched Poshan Abhiyaan program that also provide albendazole with a view towards interrupting transmission.

## Supporting information

**S1 STROBE Checklist. Strengthening the Reporting of Observational Studies in Epidemiology (STROBE) checklist.**
(DOCX)

**S1 Table. Comparison of characteristics of those consenting and refusing participation in the longitudinal monitoring cohort.**
(DOCX)

**S2 Table. Prevalence of other helminths that were detected by Kato Katz (n = 6089).**
(DOCX)

## Acknowledgments

We thank our field managers Rajeshkumar Rajendiran (Timiri) and Chinnaduraipandi (Jawadhu), study medical officer Dr. Sobana Devavaram, grant manager Samuel Paul Gideon Martin, device managers Naveen Kumar Sekar and Dhanalakshmi Manoharan, study co-ordinator Noel Joyce Mary Hillari and bio-repository manager Dhasthagir Basha for their meticulous work. We are most grateful to our field supervisors and field workers in Timiri and Jawadhu, drivers, laboratory technicians and bio-repository technicians for completing all study responsibilities in an exemplary manner. We are thankful for support extended to this study from the Ministry of Health and Family Welfare, Delhi and the Directorate of Public Health, Chennai as well as help in implementing study activities from officials at the district and block levels in both study sites. We thank the village leaders and members of our community advisory boards in Timiri and Jawadhu hills for their support. Most importantly, we thank all our cohort participants and their parents/guardians for consenting, responding to interviews and providing samples.

## Author Contributions

**Conceptualization:** Sitara S. R. Ajjampur, Katherine E. Halliday, Kristjana Ásbjörnsdóttir, Roy M. Anderson, Rajiv Sarkar, Gagandeep Kang, Judd L. Walson.

**Data curation:** Saravanakumar Puthupalayam Kaliappan, Gokila Palanisamy, Jasmine Farzana, David S. Kennedy, William E. Oswald, Sean R. Galagan.

**Formal analysis:** Saravanakumar Puthupalayam Kaliappan, Katherine E. Halliday, Gokila Palanisamy, Jasmine Farzana, William E. Oswald, Rachel L. Pullan, Sean R. Galagan, Kristjana Ásbjörnsdóttir, Roy M. Anderson, Jayaprakash Muliyil, Rajiv Sarkar.

**Funding acquisition:** Gagandeep Kang, Judd L. Walson.

**Investigation:** Sitara S. R. Ajjampur, Saravanakumar Puthupalayam Kaliappan, Gokila Palanisamy, Jasmine Farzana, Malathi Manuel, Dilip Abraham, Selvi Laxmanan, Kumudha Aruldas, Anuradha Rose, David S. Kennedy, William E. Oswald, Kristjana Ásbjörnsdóttir.

**Methodology:** Sitara S. R. Ajjampur, Saravanakumar Puthupalayam Kaliappan, Katherine E. Halliday, Gokila Palanisamy, Jasmine Farzana, Malathi Manuel, Dilip Abraham, Selvi Laxmanan, Kumudha Aruldas, Anuradha Rose, David S. Kennedy, William E. Oswald, Rachel L. Pullan, Sean R. Galagan, Kristjana Ásbjörnsdóttir, Jayaprakash Muliyil, Rajiv Sarkar.

**Project administration:** Sitara S. R. Ajjampur, Judd L. Walson.

**Resources:** Rachel L. Pullan, Gagandeep Kang, Judd L. Walson.

**Software:** Gokila Palanisamy, David S. Kennedy, William E. Oswald, Sean R. Galagan, Kristjana Ásbjörnsdóttir, Roy M. Anderson.

**Supervision:** Sitara S. R. Ajjampur, Saravanakumar Puthupalayam Kaliappan, Katherine E. Halliday, Dilip Abraham, Selvi Laxmanan, Kumudha Aruldas, Anuradha Rose, William E. Oswald, Roy M. Anderson, Jayaprakash Muliyil, Judd L. Walson.

**Validation:** Katherine E. Halliday, Malathi Manuel, Selvi Laxmanan, Sean R. Galagan.

**Visualization:** Gokila Palanisamy.

**Writing – original draft:** Sitara S. R. Ajjampur, Saravanakumar Puthupalayam Kaliappan, Katherine E. Halliday.

**Writing – review & editing:** Sitara S. R. Ajjampur, Saravanakumar Puthupalayam Kaliappan, Katherine E. Halliday, Gokila Palanisamy, Jasmine Farzana, Malathi Manuel, Dilip Abraham, Kumudha Aruldas, Anuradha Rose, David S. Kennedy, William E. Oswald, Rachel L. Pullan, Sean R. Galagan, Kristjana Ásbjörnsdóttir, Roy M. Anderson, Jayaprakash Muliyil, Rajiv Sarkar, Gagandeep Kang, Judd L. Walson.

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
