## [Decision Letter · Decision Letter 0]

11 Jan 2021

Dear Ajjampur,

Thank you very much for submitting your manuscript "Epidemiology of soil transmitted helminths and risk analysis of hookworm infections in the community: results from the DeWorm3 Trial in southern India" for consideration at PLOS Neglected Tropical Diseases. As with all papers reviewed by the journal, your manuscript was reviewed by members of the editorial board and by several independent reviewers. The reviewers appreciated the attention to an important topic. Based on the reviews, we are likely to accept this manuscript for publication, providing that you modify the manuscript according to the review recommendations. 

Sincerely,

Mar Siles-Lucas

Deputy Editor

Mar Siles-Lucas

Deputy Editor

Reviewer's Responses to Questions

**Key Review Criteria Required for Acceptance?**

**Methods**

-Are the objectives of the study clearly articulated with a clear testable hypothesis stated?

-Is the study design appropriate to address the stated objectives?

-Is the population clearly described and appropriate for the hypothesis being tested?

-Is the sample size sufficient to ensure adequate power to address the hypothesis being tested?

-Were correct statistical analysis used to support conclusions?

-Are there concerns about ethical or regulatory requirements being met?

Reviewer #1: The objectives and study design are clear. The setup seems appropriate and thorough, and the aims are reasonable given the study design. The statistical methods appear appropriate and ethical requirements appear to have been followed.

• Please provide more info about how you went from 10144 to 8517. It’s not clear to me from Figure 2 either—what made the additional people “not required”?

• Were surveys undertaken entirely in Tamil, or were any tribal or other languages used?

Reviewer #2: Yes. This study is nested within a much larger, very well-resourced and very well designed, longitudinal cohort study. It would improve with further clarity on the stool specimens collected:

Since the predominant parasite identified is hookworm, which has the least in vitro survival (once the stool is voided, from amongst the other STH parasites), and the key study outcome is intensity or number of hookworm eggs per gram of stools, it is very important to share the time from stool voiding to reading the slides. A delay can falsely underestimate the intensity. It is mentioned that stools were collected daily and transferred for lab reading every day: not clear how that translates to individual sample read time (median/mean and range, etc.).

Reviewer #3: (No Response)

**Results**

-Does the analysis presented match the analysis plan?

-Are the results clearly and completely presented?

-Are the figures (Tables, Images) of sufficient quality for clarity?

Reviewer #1: The results are clear, reasonable, and match the analysis plan. The figure and tables are appropriate and interesting. The image resolution appears a bit low, but perhaps will be fine if they are not shown in this large size in the final version.

• Your main findings in terms of correlates of hookworm infection seem to be environmental (though you acknowledge that this may be affected by environmental differences between the study sites), SES/education, and history of deworming. You use a mixture of univariable and multivariable models, but I would like to know whether history of deworming and SES/education are correlated. Two potential ways I could see this occurring are if a) people with more education had better access to healthcare or b) people with more education were more aware of past deworming as they had been provided with information about past treatment in a way that was accessible to them. If they are correlated with each other, it might be worth noting, just as you mention how you controlled for other variables when looking at social determinants. If not, then no issues.

Reviewer #2: Excellent and clearly presented results linked to the study objective. Minor question below can be clarified:

1. Authors report a higher refusal rate in persons from higher SES and education status (page 13 - Results). Could this translate to possible overestimation of the hookworm prevalence?

2. What was the reported PCT coverage for previous LF MDAs in the study areas? Would be good to know to relate to possible impact on Ascaris and Trichuris (though latter is less impacted by Albendazole treatment alone).

Reviewer #3: (No Response)

**Conclusions**

-Are the conclusions supported by the data presented?

-Are the limitations of analysis clearly described?

-Do the authors discuss how these data can be helpful to advance our understanding of the topic under study?

-Is public health relevance addressed?

Reviewer #1: The conclusions are well-supported, limitations are described (perhaps a bit more on potential bias especially in terms of the cascade of how participants were selected or opted out all the way down from the census to the number in the end with parasitological data). Good implications and relevance. Perhaps address any concerns if there's any chance (mentioned above in results comments) that people with lower educational attainment are less likely to report recent deworming (or that these two variables are potentially related in any other way).

Reviewer #2: Yes.

Reviewer #3: (No Response)

**Editorial and Data Presentation Modifications?**

Reviewer #1: • Since biannual can mean either twice a year or once every two years, please clarify by changing to “twice a year”

• I think Lo 2018 might be better summarized as something like “a study using data from 45 countries found that there was a consistent association between deworming and reduced stunting in pre-school-age children (PSAC)” rather than “multiple studies have shown the benefits of deworming in endemic countries including a reduction in stunting in pre-school-age children (PSAC)”

• Perhaps you could define an anganwadi center

• I quibble with calling school-based deworming a “highly effective approach”, as it might be interpreted as meaning highly effective at reducing infection, which is the subject of your study rather than the basis for it. You might rephrase to “this is a highly effective way of reaching PSAC and SAC”.

Reviewer #2: As above

Reviewer #3: (No Response)

**Summary and General Comments**

Reviewer #1: Overall, this is a thoughtfully done study looking at correlates of hookworm infection at two study sites, which is nested within a much larger deworming study. The correlates of reinfection are not too surprising, but this work is important. The finding that hookworm prevalence remains high is more interesting than it would otherwise be, as the authors have extensive knowledge of past community-based deworming for LF and school-based deworming in these study areas. Thus, the high prevalence is not just a snapshot of a random location, but proof that the programs which have targeted these areas have not yet achieved the elimination of STH as a public health problem in these areas.

Reviewer #2: This is a very well designed, well-conducted study which will add important knowledge for STH control (underscoring the confirmation of the need to treat adults as well as children to holistically address STH morbidity, especially in areas with a high burden of hookworm).

Reviewer #3: (No Response)

PLOS authors have the option to publish the peer review history of their article (what does this mean?). If published, this will include your full peer review and any attached files.

Reviewer #1: No

Reviewer #2: Yes: Rubina Imtiaz

Reviewer #3: Yes: Luc E. Coffeng
---

## [Decision Letter · Decision Letter 1]

29 Mar 2021

Dear Ajjampur,

We are pleased to inform you that your manuscript 'Epidemiology of soil transmitted helminths and risk analysis of hookworm infections in the community: results from the DeWorm3 Trial in southern India' has been provisionally accepted for publication in PLOS Neglected Tropical Diseases.

Best regards,

Mar Siles-Lucas

Deputy Editor

Mar Siles-Lucas

Deputy Editor

Reviewer's Responses to Questions

**Key Review Criteria Required for Acceptance?**

**Methods**

-Are the objectives of the study clearly articulated with a clear testable hypothesis stated?

-Is the study design appropriate to address the stated objectives?

-Is the population clearly described and appropriate for the hypothesis being tested?

-Is the sample size sufficient to ensure adequate power to address the hypothesis being tested?

-Were correct statistical analysis used to support conclusions?

-Are there concerns about ethical or regulatory requirements being met?

Reviewer #1: (No Response)

Reviewer #3: (No Response)

**Results**

-Does the analysis presented match the analysis plan?

-Are the results clearly and completely presented?

-Are the figures (Tables, Images) of sufficient quality for clarity?

Reviewer #1: (No Response)

Reviewer #3: (No Response)

**Conclusions**

-Are the conclusions supported by the data presented?

-Are the limitations of analysis clearly described?

-Do the authors discuss how these data can be helpful to advance our understanding of the topic under study?

-Is public health relevance addressed?

Reviewer #1: (No Response)

Reviewer #3: (No Response)

**Editorial and Data Presentation Modifications?**

Reviewer #1: (No Response)

Reviewer #3: (No Response)

**Summary and General Comments**

Reviewer #1: I feel that the responses to my suggestions have been comprehensive.

Reviewer #3: (No Response)

PLOS authors have the option to publish the peer review history of their article (what does this mean?). If published, this will include your full peer review and any attached files.

Reviewer #1: No

Reviewer #3: **Yes: **Luc E. Coffeng

---

## [Editor Report · Acceptance letter]

27 Apr 2021

Dear Ajjampur,

We are delighted to inform you that your manuscript, "Epidemiology of soil transmitted helminths and risk analysis of hookworm infections in the community: results from the DeWorm3 Trial in southern India," has been formally accepted for publication in PLOS Neglected Tropical Diseases.

Best regards,

Shaden Kamhawi

co-Editor-in-Chief

Paul Brindley

co-Editor-in-Chief
